# Citizen science with colour blindness: A case study on the Forel-Ule scale

Olivier Burggraaff[1,2]*, Sanjana Panchagnula[1,3], Frans Snik[1]

**1** Leiden Observatory, Leiden University, Leiden, The Netherlands, **2** Institute of Environmental Sciences (CML), Leiden University, Leiden, The Netherlands, **3** Laboratory for Astrophysics, Leiden Observatory, Leiden University, Leiden, The Netherlands

* burggraaff@strw.leidenuniv.nl

**Data Availability Statement:** The data underlying this study are from Novoa et al. 2013 (http://dx.doi.org/10.2971/jeos.2013.13057). All findings can be entirely replicated using the data from Novoa et al. 2013 and the protocol in our Methods section. The

## Abstract

Many citizen science projects depend on colour vision. Examples include classification of soil or water types and biological monitoring. However, up to 1 in 11 participants are colour blind. We simulate the impact of various forms of colour blindness on measurements with the Forel-Ule scale, which is used to measure water colour by eye with a 21-colour scale. Colour blindness decreases the median discriminability between Forel-Ule colours by up to 33% and makes several colour pairs essentially indistinguishable. This reduces the precision and accuracy of citizen science data and the motivation of participants. These issues can be addressed by including uncertainty estimates in data entry forms and discussing colour blindness in training materials. These conclusions and recommendations apply to colour-based citizen science in general, including other classification and monitoring activities. Being inclusive of the colour blind increases both the social and scientific impact of citizen science.

## 1 Introduction

Colour measurements are common in citizen science. They are often done using red-green-blue (RGB) consumer cameras such as smartphones [1–3], but also with the human eye. Human colour measurements are used in such diverse fields as coral reef monitoring [4], snail evolution [5], soil surveying [6], climate adaptation [7], and water colour [8–10]. The data are expressed through a qualitative label [5] or by comparison with a colour chart [4, 6, 8–11]. Colour is a useful proxy for underlying properties such as chemical composition [11, 12] and the simplicity of measuring with the eye enables low-cost measurements over large areas and long time series [12, 13].

Accessibility and inclusivity are key to successful citizen science [14, 15]. A large and diverse group of participants increases the social and scientific impact of citizen science [5, 14, 16, 17]. However, recruiting and retaining participants is challenging [14, 15, 18, 19]. Important motivations to participate are a feeling of contributing to science and environmental protection [6, 16–19], learning [7, 17, 18], and simply having fun [9, 14, 16, 17, 19]. Common reasons to stop participating include mis- or not understanding the project [17], perceiving

authors did not have any special access privileges that others would not have.

**Funding:** This project has received funding from the European Union's Horizon 2020 research and innovation programme under grant agreement No 776480. The funders had no role in study design, data collection and analysis, decision to publish, or preparation of the manuscript.

**Competing interests:** The authors have declared that no competing interests exist.

the data as not valuable [4, 16, 18], and difficulty in performing the measurements [6, 9, 18, 20].

While colour vision is often assumed to be universal, many differences exist between individuals. Colour blindness, or colour deficiency, affects up to 9% of men and 2% of women, depending on ethnicity and other genetic factors [21]. It reduces or even eliminates one's ability to distinguish certain colours, most commonly red and green [21]. Colour blindness is typically congenital [21–25], but can also be acquired through age or disease [25, 26].

Three forms of colour blindness exist, namely *anomalous trichromacy*, *dichromacy*, and *monochromacy*. Each affects the eye's three pigments in a different way. These pigments are labelled LMS for long-, medium-, and short-wave, respectively, with peak sensitivity wavelengths of 560, 530, and 420 nm [22]. In anomalous trichromacy, a single pigment has an atypical spectral response, reducing one's colour discrimination abilities [22, 23, 27]. This is called *protanomaly*, *deuteranomaly*, or *tritanomaly*, for the respective LMS pigments. Dichromacy is a complete lack of one pigment, similarly called *protanopia*, *deuteranopia*, or *tritanopia* [22]. Finally, monochromacy is a complete lack of multiple cones, causing a full lack of colour vision. Monochromacy is exceedingly rare [22, 24] and is not discussed further in this work.

Colour blindness is often treated as a continuous spectrum from regular colour vision (all pigments present and typical) through degrees of anomalous trichromacy (one pigment atypical) to dichromacy (one pigment wholly missing) [21, 22, 26]. For simplicity, the three LMS deficiencies are referred to as *protan*, *deutan*, and *tritan*, respectively [21]. Protan and deutan are the most common, affecting for example up to 9% of men and 0.6% of women in Europe, as well as 7% of men and 2% of women in China [21]. The prevalence of tritan in the West is on the order of 1:10 000 [25], though higher prevalences have been reported in other locations [28].

Colour blindness limits the accessibility of citizen science that involves colour measurements for up to 1 in 11 participants. However, to our knowledge, little research has gone into its potentially far-reaching consequences. Such work has been done for science communication, for example in designing inclusive colour maps [27, 29].

As a case study, we investigate the impact of colour blindness on water colour measurements with the Forel-Ule (FU) scale. This scale quantifies human water colour measurements [30] by assigning a numerical value from 1–21 to a predetermined set of colours, shown in Fig 1. These range from indigo blue (FU 1) through green (FU 11) to cola brown (FU 21). First used in the 1890s by Forel and Ule [31, 32], it provides the longest continuous record of ocean colour [13]. For instance, Wernand and Van der Woerd used 17 171 archival FU measurements from 1930 to 1999 to derive long-term biogeochemical trends in the Pacific Ocean [12]. Properties of a water body that can be derived from its colour include suspended particles, dissolved organic matter, and algal pigments such as chlorophyll-a [12, 13, 33].

The FU scale is commonly used by professionals [12, 34] and by citizen scientists [8, 10]. Measurements are done by comparing a physical standard colour scale to a water body. For citizen science, the original scale made from 21 vials of pigment mixes [35] may be replaced with plastic filters [10] or a printed version [8], making it easier to use. Having this physical reference reduces the effects of variations in illumination, though in all cases it is difficult to guarantee colour consistency.

We use simulations to determine the effects of colour blindness on FU measurements. Such digital simulations accurately reproduce colour blind vision [27, 36]. The discriminability of the resulting shifted colours is assessed using the CIE $\Delta E_{00}$ colour difference measure [37]. This way, the impact of colour blindness on FU measurements is quantified.

Based on these results, we make general recommendations for dealing with colour blindness in citizen science. These include guidelines for data entry protocols and training

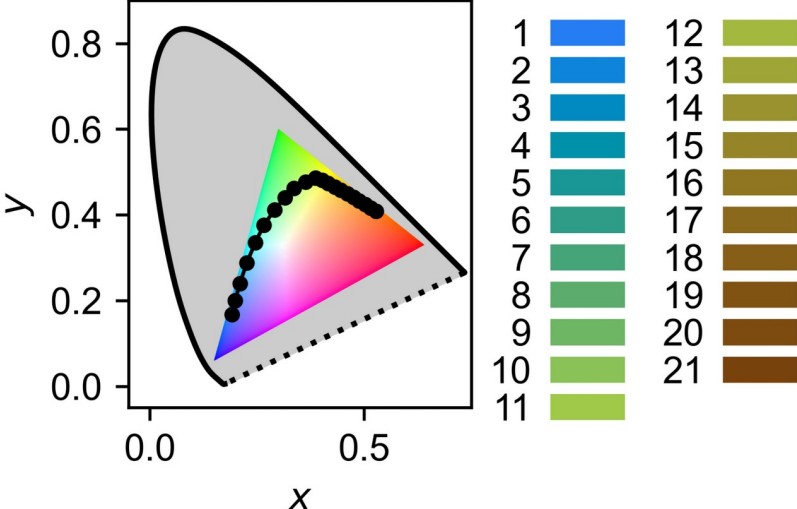

**Fig 1. The Forel-Ule scale.** The individual FU colours are shown on the right, a comparison to the human gamut on the left. The gamut is plotted in $(x, y)$ chromaticity, normalized from CIE XYZ and shown with a constant brightness, and converted to sRGB. The FU scale increases from 1 (bottom left) to 21 (far right). The shaded area represents the full gamut of regular colour vision, while the coloured triangle represents the sRGB colour space, which most computer monitors are limited to. The perceived colours may vary depending on monitor or printer settings and the reader's own colour vision.

materials, benefiting citizen motivation and data quality. Moreover, the methods applied in this work are easily generalized to other colour-based tools. This enables authors to account for colour blindness in the design stage of new citizen science projects. While some projects have opted for simplified colour scales [11], this significantly reduces the information content [33] of all data, including those from colour blind participants. Simplified colour scales are thus generally not an ideal solution to this problem.

Section 2 describes the methods used to simulate colour deficiency and assess colour discriminability. Results are presented in Section 3 and discussed in Section 4. Finally, conclusions and recommendations are drawn up in Section 5.

## 2 Methods

The colour blindness simulations and analysis were implemented in custom Python scripts available from https://github.com/burggraaff/cbfu.

### 2.1 Forel-Ule scale

Tristimulus (CIE XYZ) values for the FU scale were derived by Novoa et al. from transmission spectroscopy [35]. The corresponding $(x, y)$ chromaticities are shown in Fig 1.

Four illuminants were considered, namely E (equal-energy) and D55, D65, and D75 (daylight). These illuminants quantify differences in lighting conditions and are used to express colour appearance in a standardised manner [38]. The FU scale is defined with an E illuminant [35] but measurements take place in daylight, making D-type illuminants more representative [10]. Conversion between illuminants was done in XYZ space using the Bradford chromatic adaptation matrices provided on Bruce Lindbloom's website [39].

The tristimulus values were first converted to the LMS colour space, representing the relative excitations of the LMS cones [27, 40]. This was done through the Hunt-Pointer-Estevez matrix [40], as shown in Eq (1). Here $[L\,M\,S]^T$ and $[X\,Y\,Z]^T$ are the vector representations of a

single colour in LMS and XYZ, respectively.

$$\begin{bmatrix} L \\ M \\ S \end{bmatrix} = \begin{bmatrix} 0.38971 & 0.68898 & -0.07868 \\ -0.22981 & 1.18340 & 0.04641 \\ 0 & 0 & 1.00000 \end{bmatrix} \begin{bmatrix} X \\ Y \\ Z \end{bmatrix} \tag{1}$$

## 2.2 Simulation of colour blindness

Colour blindness was simulated by mapping colours from the LMS colour space representing regular vision to a reduced colour space representing colour deficiency [27, 36, 41]. This is a mathematical representation of how colour appearances shift due to colour blindness, based on the observed colour perceptions of dichromats [36]. Since for dichromats and anomalous trichromats, two out of three cones are unaffected, the responses of those cones to a given colour are unchanged. The simulation determines the response of the third, deficient cone that imitates for a regular observer the colour perceived by a colour blind person [36, 41]. This in turn allows us to apply discriminability metrics developed for regular colour vision to the simulated perceived colours.

The LMS-space vectors $\vec{c}_L$ were modified using a cone-deficiency transfer matrix $\mathbf{T_k}$. $\mathbf{T_k}$ is the identity matrix $\mathbf{I_3}$ with one diagonal element ($T_k^{00}$, $T_k^{11}$, $T_k^{22}$ for protan, deutan, tritan, respectively) reduced to a relative cone contribution $k$. This is shown in Eqs (2) and (3) for protan with its respective matrix $\mathbf{T_k^p}$ [27, 41]. $k$ ranges continuously from 1 (regular vision) to 0 (dichromacy). It represents the relative contribution of a specific cone to colour vision but does not correspond directly to a physical property of the eye. The elements $q_1$ and $q_2$ of $\mathbf{T_k}$ shift the response from the deficient cone (L in the example) to the others.

$$\begin{bmatrix} L' \\ M' \\ S' \end{bmatrix} = \begin{bmatrix} k & q_1^p & q_2^p \\ 0 & 1 & 0 \\ 0 & 0 & 1 \end{bmatrix} \begin{bmatrix} L \\ M \\ S \end{bmatrix} \tag{2}$$

$$\vec{c}_L' = \mathbf{T_k^p} \vec{c}_L \tag{3}$$

The cone transfer matrices for protan $\mathbf{T_k^p}$, deutan $\mathbf{T_k^d}$, and tritan $\mathbf{T_k^t}$ are as follows:

$$\mathbf{T_k^p} = \begin{bmatrix} k & q_1^p & q_2^p \\ 0 & 1 & 0 \\ 0 & 0 & 1 \end{bmatrix} \quad \mathbf{T_k^d} = \begin{bmatrix} 1 & 0 & 0 \\ q_1^d & k & q_2^d \\ 0 & 0 & 1 \end{bmatrix} \quad \mathbf{T_k^t} = \begin{bmatrix} 1 & 0 & 0 \\ 0 & 1 & 0 \\ q_1^t & q_2^t & k \end{bmatrix} \tag{4}$$

The elements $q_1$, $q_2$ were determined by noting that colour blind people retain regular vision for white and a complementary colour (blue for protan and deutan, red for tritan) [27, 36, 41]. In other words, $\mathbf{T_k}$ has eigenvectors $\vec{w}_L = [1\,1\,1]^T$ (white) and either $\vec{b}_L$ (blue) or $\vec{r}_L$

(red) with eigenvalues 1. This is shown in Eqs (5) and (6).

$$\mathbf{T_k^p}\vec{b}_L \; = \vec{b}_L \quad \mathbf{T_k^d}\vec{b}_L \; = \vec{b}_L \quad \mathbf{T_k^t}\vec{r}_L \; = \vec{r}_L \tag{5}$$

$$\mathbf{T_k^p}\vec{w}_L \; = \vec{w}_L \quad \mathbf{T_k^d}\vec{w}_L \; = \vec{w}_L \quad \mathbf{T_k^t}\vec{w}_L \; = \vec{w}_L \tag{6}$$

For each case, a system of two equations with two unknowns $q_1$, $q_2$ and one variable $k$ was derived, with $L_b$, $M_b$, $S_b$ the LMS coordinates of the blue reference vector $\vec{b}_L$ and $L_r$, $M_r$, $S_r$ those of $\vec{r}_L$:

$$kL_b + q_1^p M_b + q_2^p S_b \; = L_b \quad kM_b + q_1^d L_b + q_2^d S_b \; = M_b \quad kS_r + q_1^t L_r + q_2^t M_r \; = S_r \tag{7}$$

$$k + q_1^p + q_2^p \; = 1 \quad k + q_1^d + q_2^d \; = 1 \quad k + q_1^t + q_2^t \; = 1 \tag{8}$$

Solving for $q_1$, $q_2$ gave the following expressions:

$$q_1^p \; = 1 - k - q_2^p \quad q_1^d \; = 1 - k - q_2^d \quad q_1^t \; = 1 - k - q_2^t \tag{9}$$

$$q_2^p \; = (1-k)\frac{M_b - L_b}{M_b - S_b} \quad q_2^d \; = (1-k)\frac{L_b - M_b}{L_b - S_b} \quad q_2^t \; = (1-k)\frac{L_r - S_r}{L_r - M_r} \tag{10}$$

The sRGB blue and red primaries are typically used for $\vec{b}_L$ and $\vec{r}_L$, respectively, as this technique is used in the field of computer graphics [27, 41]. While other primaries could be used, such as monochromatic wavelengths [36], this makes little difference [27] so we followed the convention.

We calculated $\mathbf{T_k}$ for protan, deutan, and tritan with $1 \geq k \geq 0$ in intervals of 0.01, and transformed the 21 FU colours with each $\mathbf{T_k}$. The modified vectors were then transformed back to XYZ and analyzed. This was implemented in Python through NumPy's `einsum` method [42].

## 2.3 Colour discrimination

Discriminability of the transformed FU colours was assessed in the CIE Lab (1976) colour space. CIE Lab is approximately perceptually uniform, its components representing lightness (L*), green-red (a*), and blue-yellow (b*) [38]. While FU colour assignment is typically done in $(x, y)$ chromaticity (normalized XYZ) through the hue angle [3, 33], this approach does not work for dichromacy, which reduces the chromaticity plane to a line [36]. The Euclidean distance in XYZ coordinates also could not be used, as XYZ is not perceptually uniform [43].

Discriminability was quantified through the $\Delta E_{00}$ metric [37], which expresses the difference between colour pairs. The full formula for $\Delta E_{00}$ is given in [37] and not reprinted here due to its length; our Python implementation passed all the example cases in said paper. A value of $\Delta E_{00} = 2.3$ corresponds to a just-noticeable difference (JND), the smallest difference an average observer can distinguish [38, 44].

For each deficiency simulation, the $\Delta E_{00}$ difference between each of the 21 transformed FU colours was calculated, giving a $21 \times 21$ confusion matrix. In this, any colour pairs where $\Delta E_{00} < 1$ JND cannot be discriminated at all, while pairs with $1 \leq \Delta E_{00} \leq 3$ are discriminable with difficulty. Pairs with $\Delta E_{00} > 3$ were considered discriminable.

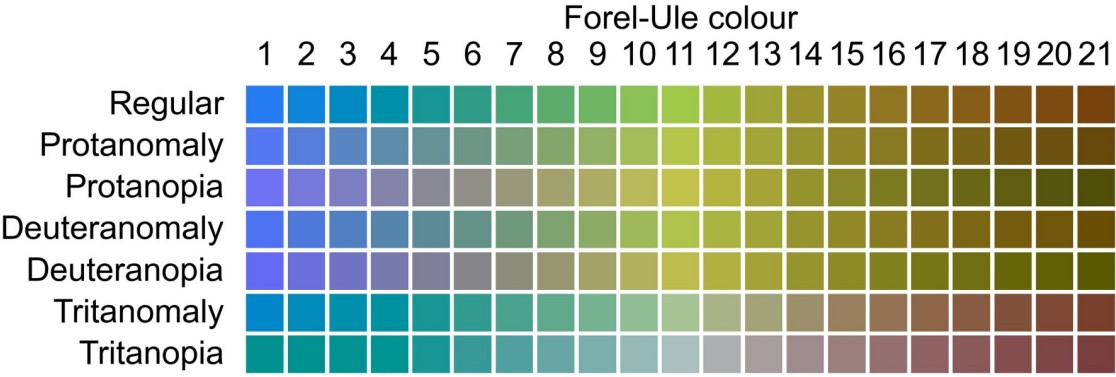

**Fig 2. Apparent Forel-Ule colours with regular and deficient colour vision.** The (modified) XYZ coordinates were adapted to a D65 illuminant, then converted to the sRGB colour space and gamma expanded [1] for visualization purposes. The perceived colours may vary depending on monitor or printer settings and the reader's own colour vision. Readers who cannot distinguish between the colours shown here may benefit from taking a colour vision test; many variants are freely available online. The anomalous examples correspond to $k = 0.50$.

## 3 Results

### 3.1 Colour blindness simulation

The appearance of the FU scale with varying degrees of colour blindness, simulated as in Section 2.2, is shown in Fig 2. The observed changes qualitatively match those seen in previous work [27, 41] and were anecdotally confirmed by one of the authors (deuteranomalous) and a colleague (protanopic). The largest colour shifts are seen for tritan, as expected since it affects the perception of blue light and many FU colours are shades of blue.

Colour blindness narrows the gamut of the FU scale, as shown in Fig 3. It has little effect on the lightness ($L^*$) of the FU scale but affects its colour components. Protan and deutan (red-green blindness) reduce the range of $a^*$ (red-green) while tritan reduces the range of $b^*$ (blue-yellow). These shifts imply that colour blindness reduces the ability to discriminate FU colours based on hue, meaning the user will have to rely more on lightness.

### 3.2 Colour discrimination

The discriminability of FU colours is reduced by colour blindness. The confusion matrices for regular and deficient vision, calculated as in Section 2.3, are shown in Fig 4. They show that the reduced range in $a^*$ (red-green) for protan and deutan and in $b^*$ (blue-yellow) for tritan, observed in Section 3.1, reduce the discriminability at opposite ends of the FU scale. The former primarily affect FU 10–21 (green–brown) while tritan affects FU 1–9 (blue–green).

Several pairs of FU colours become fully indistinguishable. Deuteranopia causes two colour pairs (FU 19-20 and 20-21) to fall within 1 JND and thus become indistinguishable. For tritanopia, six pairs become indistinguishable, namely 1-2, 1-3, 2-3, 3-4, 4-5, and 5-6. Protanopia does not cause indistinguishable pairs.

Additionally, many more pairs exhibit reduced discriminability. While most adjacent pairs are <3 JND apart even with regular colour vision, deficiency extends this further off the diagonal. In particular, protan and deutan cause confusion between the central colours (FU 9–13), which is also apparent from Fig 3 as they have similar $L^*$, $a^*$, and $b^*$. On the other hand, tritan significantly reduces the discriminability of FU 1–9. As seen in Fig 5, the number of pairs within 3 JND increases from 17 (regular) to 24 (protanopia), 21/24 (deuteranopia/deuteranomaly), or 30 (tritanopia).

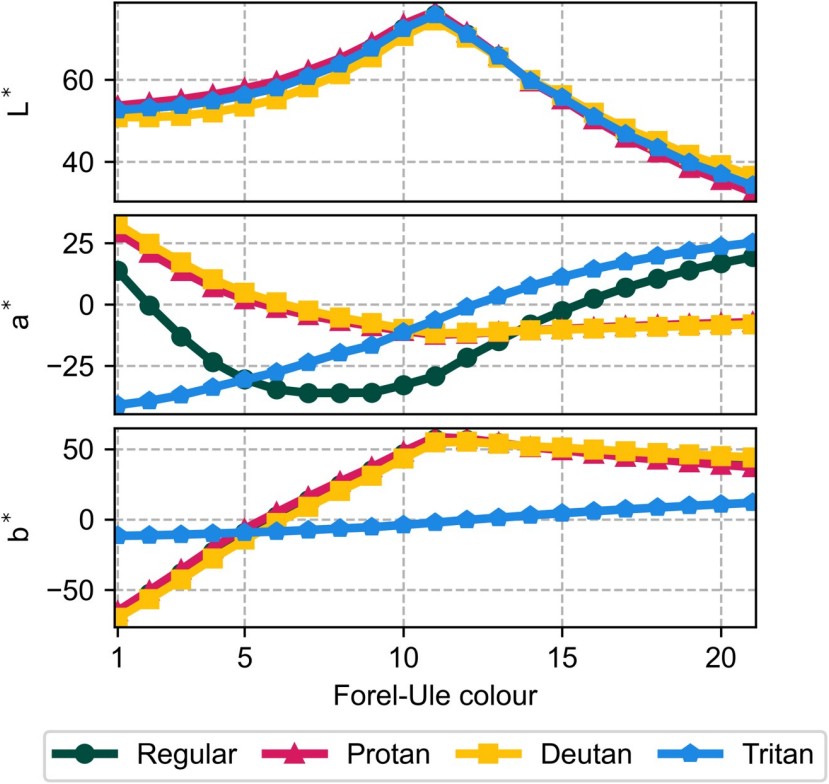

**Fig 3. Forel-Ule colours in CIE Lab space.** Both regular and deficient vision are included. Regular vision is hidden in the top and bottom panels behind protan and deutan. These affect the a* (green-red) coordinate the most while tritan affects b* (blue-yellow) the most. None of the deficiencies significantly affect L* (lightness).

These trends also apply to partial colour blindness (anomalous trichromacy). Fig 5 shows the relation between $k$ and median/minimum $\Delta E_{00}$ as well as the number of indistinguishable pairs. The median decreases smoothly for protan, deutan, and tritan (from 33 to 27, 26, and 22, respectively) from $k = 1$ to 0. The minimum $\Delta E_{00}$ decreases smoothly for protan and deutan (from 3.3 to 2.5 and 2.2, respectively) while the tritan curve is piecewise smooth. Fully indistinguishable pairs ($\Delta E_{00} < 2.3$) appear at $k \leq 0.20$ for deutan and tritan.

Chromatic adaptation with a daylight illuminant (Section 2.1) did not affect these results. While the $\Delta E_{00}$ between some pairs changed by up to 1 JND, the patterns seen in Figs 4 and 5 remained, as did the previously discussed pairs of non-discriminable colours.

## 3.3 Practical consequences

In practice, FU measurements always have an uncertainty of $\geq 1$ FU units. This is due to viewing conditions at the time of measurement including waves, specular reflections, and uneven illumination. As seen in Section 3.2, adjacent pairs of FU colours are difficult to distinguish ($\Delta E_{00} < 3$ JND) even with regular vision.

Colour blindness increases the uncertainty on FU measurements. Observers with protan or deutan experience increased difficulty in distinguishing adjacent pairs. Moreover, protans have difficulty distinguishing FU 9–13 while for deutans, FU 19-20 and 20-21 are completely indistinguishable. For a FU 20-type water body, a deutan cannot specify their observation more precisely than 19–21. Furthermore, $\Delta E_{00} = 2.33$ for FU 18 and 20, further reducing this

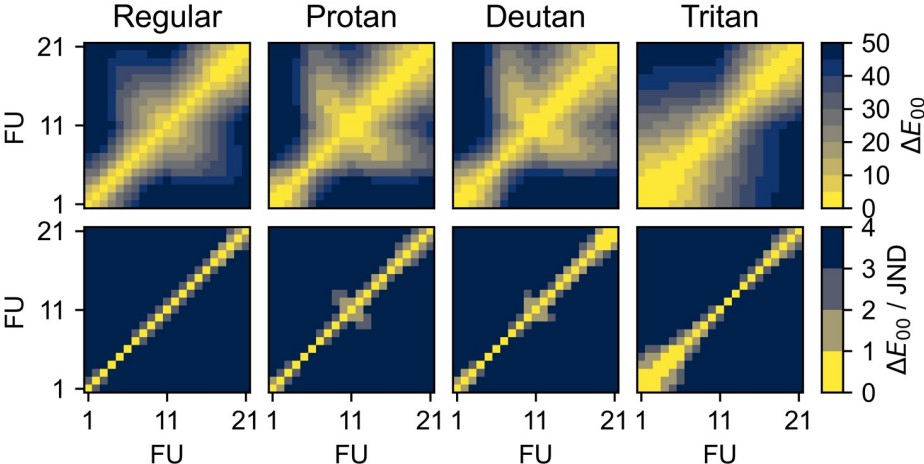

**Fig 4. Confusion matrices for regular and deficient colour vision.** The top panels show the full range of $\Delta E_{00}$, while the bottom panels have a narrower colour bar, in units of just-noticeable difference (JND, $\Delta E_{00} = 2.3$). Even with regular vision, some pairs of FU colours are difficult to distinguish ($\Delta E_{00} \leq 3$ JND) Protan and deutan primarily decrease the discriminability of the middle (green) and high (brown) colours, while tritan primarily affects the low (blue) colours, as expected.

precision to 18–21 given imperfect viewing conditions. Similarly, since tritans cannot distinguish six pairs of colours in the FU 1–6 range, they can provide little precision on 99% of global surface waters [33].

This increased uncertainty affects data quality and user motivation. This is further discussed in Section 4 and recommended guidelines for considering these issues are given in Section 5.

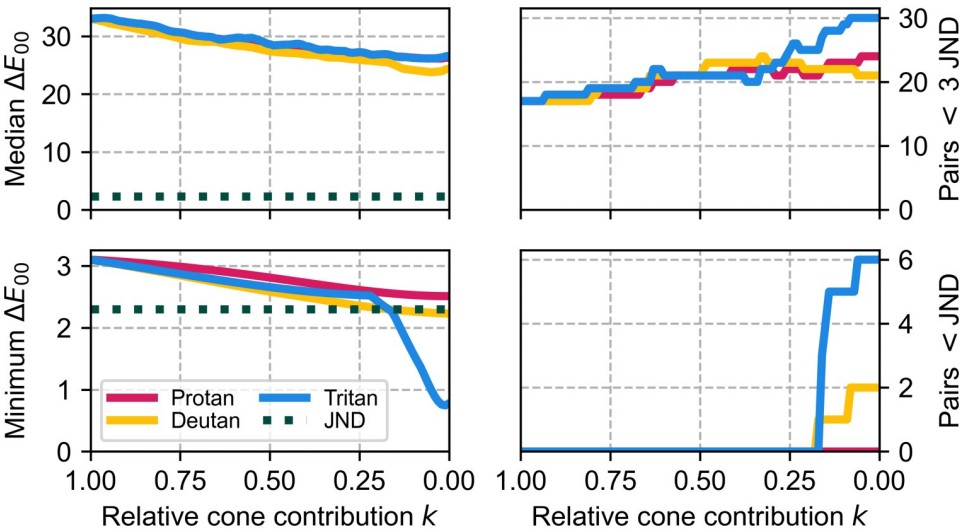

**Fig 5. Discriminability of Forel-Ule colours.** The median and minimum (left) $\Delta E_{00}$ difference between FU colour pairs, and the number of pairs within 3 and 1 JND (right), are shown as a function of the relative cone contribution $k$. $k$ ranges from 1 (full colour vision) to 0 (dichromacy), with intermediate values representing partial colour blindness (anomalous trichromacy). Pairs with $\Delta E_{00} < 1$ JND are fully indistinguishable, pairs with $<3$ JND are difficult to distinguish (Section 3.2).

## 4 Discussion

Simulating the effects of colour blindness on Forel-Ule (FU) measurements, we have found significant reductions in colour discriminability and hence precision (Sections 3.2 and 3.3). This matches the authors' and colleagues' experiences in the field, and the simulation methods are well-attested in other contexts [27, 29, 36]. However, wider validation specific to the FU scale, with participants representing different types of colour blindness, is desirable.

The reduction in precision due to colour blindness reduces the quality and value of citizen science data. The magnitude of this effect depends on the type and severity of colour blindness, as described in Section 3.3. Protans and deutans, the vast majority of colour blind people in the West [21, 25], experience a reduction in median discriminability ($\Delta E_{00}$) between FU colours of up to 21%; for tritans this is 33%. The uncertainty in FU data increases correspondingly, though not evenly. For example, tritans' ability to identify green-brown waters (FU 10–21) changes little, but they cannot distinguish the blue water types (FU 1–6) that represent most global surface waters [33].

This reduction in data value can be addressed by modifying data entry protocols to include uncertainties. Currently, many citizen science projects require users to provide a single value, for example FU 9 or 10. An entry field for uncertainty, or allowing the user to enter multiple values, accounts for the decrease in selectivity. Participants can estimate this uncertainty themselves. Even FU measurements by participants with regular colour vision have a typical uncertainty of ±1 FU (Section 3.2), which should be accounted for when using them to validate remote sensing data [34]. Colour blindness, particularly dichromacy, increases this uncertainty to up to ±3 FU.

We propose three methods to include uncertainties in data entry forms. The first is simply to include two fields, one for the best estimate (for example FU 9) and one for the estimated uncertainty (e.g. ±2 FU). This method is commonly used in scientific publications but it may be difficult for citizens to understand and apply [45], especially for asymmetric uncertainties. The second method is to have participants estimate a sequential range of possible values (e.g. FU 8–11), optionally including a single best estimate (e.g. FU 9). This is intuitive, simple to apply, and easily translated into traditional uncertainty intervals. It is most applicable for sequential scales like FU where confusion occurs primarily between adjacent numbers (Fig 4). The third method is to have participants select any number of possible values (e.g. FU 8, 9, 11). This is the most general method for discrete colour scales but makes the uncertainties more difficult to process. It is best suited to colour scales with many non-adjacent indistinguishable pairs. Our Python code (Section 2) can be adapted to other colour scales to determine which method is most suitable. A more detailed discussion on handling uncertainty in citizen science data is provided in [46].

Colour blindness can also affect the motivation of citizen scientists. As discussed in Section 1, participants need to feel they are contributing to science with valuable data. A participant presented with a colour scale where multiple colours appear indistinguishable may dismiss the method as either too difficult or nonsensical, and stop participating [4, 6, 9, 16–18, 20]. This is especially true for one unaware of their colour blindness. Since citizen science benefits from a large and diverse group of participants [5, 14, 16, 17], participant retention is important.

Demotivation can be prevented by modifying training materials. Explaining the choice of colour scale and how colour blindness affects its appearance helps participants understand the method. Particular care should be taken in emphasising the value of citizen data, even with colour blindness. For example, while tritans cannot distinguish the FU colours covering the open sea, their ability to distinguish FU 10–21 differs little from regular vision. These cover many

inland waters [3, 47], which are commonly studied with the FU scale [30], so training materials should emphasise the value of tritans' observations there.

Training participants to estimate and provide uncertainties would further help them understand the value of their data [45]. Moreover, since uncertainty estimation is an integral part of professional science, citizen scientists may even gain motivation from learning about it [7, 17, 18]. For existing applications, if modifying data entry forms is impossible, explaining why colours may appear similar and how to pick a single colour would reduce the perceived difficulty.

The severity of these motivational effects and the efficacy of these preventative measures should be tested in practice. Comparing the retention of participants with regular and deficient colour vision, with and without modified training materials and data entry forms, would serve this purpose. This is ideally done in the design stage, as part of a co-creation process [7, 17].

Additional future work includes investigating the effects of other variations in colour perception. Even among those with regular colour vision, variations in colour perception exist [22], including demographical trends [21, 28]. Moreover, monochromacy was not discussed in this work because of its rarity [24] but likely has an even more pronounced effect on colour discriminability than the deficiencies investigated here.

Finally, unrelated to human observations, Fig 3 highlights the importance of lightness in distinguishing FU colours. Many FU index algorithms, which apply the FU scale to remote sensing data, only account for chromaticity [3, 33, 34]. Introducing lightness to these algorithms may improve their precision and accuracy.

## 5 Conclusions & recommendations

Citizen science projects that depend on colour vision should account for colour blindness, which affects up to 1 in 11 participants. For Forel-Ule water colour measurements, colour blindness reduces the median discriminability between colours by up to 33% and makes multiple pairs of colours fully indistinguishable. This affects data quality and citizen motivation.

Modifying data entry forms to include uncertainty estimates would reduce the impact on data quality. This can be done by letting participants estimate the uncertainty in their measurement or choose multiple colours on the scale. Our provided Python code can be adapted to determine the best suited method for different colour scales. Learning how to estimate uncertainties may also increase participants' motivation and understanding of science.

The impact on motivation is reduced by including colour blindness in training materials. This includes explaining the colour scale and the difficulties colour blind participants may face, but also emphasising the continued value of their data. Through improved retention, this increases the number and diversity of the participants, which in turn increases both the social and scientific impact of citizen science.

## Acknowledgments

The authors wish to thank Mortimer Werther, Steele Farnsworth, Emmanuel Boss, and Akupara Panchagnula for valuable discussions relating to this work. Data analysis and visualization were done using the Matplotlib, NumPy, and colorio libraries for Python. Fig 4 uses the *cividis* colour map [29]; the line colours in Figs 3 and 5 were obtained from https://davidmathlogic. com/colorblind/.

## Author Contributions

**Conceptualization:** Olivier Burggraaff, Sanjana Panchagnula.

**Data curation:** Olivier Burggraaff.

**Formal analysis:** Olivier Burggraaff.

**Funding acquisition:** Frans Snik.

**Investigation:** Olivier Burggraaff, Sanjana Panchagnula.

**Methodology:** Olivier Burggraaff, Frans Snik.

**Software:** Olivier Burggraaff.

**Visualization:** Olivier Burggraaff, Sanjana Panchagnula, Frans Snik.

**Writing – original draft:** Olivier Burggraaff, Sanjana Panchagnula.

**Writing – review & editing:** Olivier Burggraaff, Sanjana Panchagnula, Frans Snik.

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
