## [Decision Letter · Decision Letter 0]

9 Mar 2021

PONE-D-21-04836

Citizen science with colour blindness: A case study on the Forel-Ule scale

PLOS ONE

Dear Dr. Burggraaff,

Thank you for submitting your manuscript to PLOS ONE. After careful consideration, we feel that it has merit but does not fully meet PLOS ONE’s publication criteria as it currently stands. Therefore, we invite you to submit a revised version of the manuscript that addresses the points raised during the review process.

I have so far had the manuscript reviewed by one expert in the field, and I have also carefully read the manuscript which is close to my research area. Reviewer 1 sees a lot of value in the study, and has provided a couple of points that would likely enhance understanding for the broad readership pf PLoS1. If you can make these revisions I will look at the paper again in detail and it should be possible to proceed.

We look forward to receiving your revised manuscript.

Kind regards,

Adrian G Dyer, Ph.D.

Academic Editor

PLOS ONE

Journal Requirements:

Reviewers' comments:

Reviewer's Responses to Questions

**Comments to the Author**

1. Is the manuscript technically sound, and do the data support the conclusions?

Reviewer #1: Yes

2. Has the statistical analysis been performed appropriately and rigorously? 

Reviewer #1: Yes

3. Have the authors made all data underlying the findings in their manuscript fully available?

Reviewer #1: Yes

4. Is the manuscript presented in an intelligible fashion and written in standard English?

Reviewer #1: Yes

5. Review Comments to the Author

Reviewer #1: Burggraaff et al present an interesting study evaluating the effect of various known conditions of colour vision anomalies on colour evaluation of RGB images as those used for characterising water quality using the Forel-Ule scale.

In my opinion the authors present the results of a well-designed experiment and based their conclusions on the observed data. I think this manuscript is suitable for publication in PlosOne after the authors have address the two minor points detailed below.

##SPECIFIC COMMENTS ##

1. Subsection 2.2 presents details of the calculations necessary to simulate the effect of the various colour deficiency conditions on the discriminability of the samples present in the FU scale. However, readers can benefit from an early introduction to the rationale of this mapping perhaps in the Introduction section or early in section 2.2. This can be done by discussing the methods by Brettel et al (1997) on which the simulations are based.

2.The authors propose modifying current data entry protocols to include measurements of uncertainty associated with reference similarity for normal and deficient colour vision (lines 217-220). This is a very interesting idea and useful conclusion of the paper. I recommend to expand on this idea based on the presented results. By doing that this manuscript will became a very useful tool for designing new protocols improving not only data collected but also its efficiency,

6. PLOS authors have the option to publish the peer review history of their article (what does this mean?). If published, this will include your full peer review and any attached files.

Reviewer #1: No

---

## [Author Response · Author response to Decision Letter 0]

18 Mar 2021

Please see our attached response to the reviewers for a full response.

---

## [Editor Report · Decision Letter 1]

25 Mar 2021

Citizen science with colour blindness: A case study on the Forel-Ule scale

PONE-D-21-04836R1

Dear Dr. Burggraaff,

We’re pleased to inform you that your manuscript has been judged scientifically suitable for publication and will be formally accepted for publication once it meets all outstanding technical requirements.

Kind regards,

Adrian G Dyer, Ph.D.

Academic Editor

PLOS ONE
---

## [Editor Report · Acceptance letter]

8 Apr 2021

PONE-D-21-04836R1 

Citizen science with colour blindness: A case study on the Forel-Ule scale 

Dear Dr. Burggraaff:

I'm pleased to inform you that your manuscript has been deemed suitable for publication in PLOS ONE. Congratulations! Your manuscript is now with our production department. 

Kind regards, 

on behalf of

Dr. Adrian G Dyer 

Academic Editor

PLOS ONE